# Baseline Presence of NAFLD Predicts Weight Loss after Gastric Bypass Surgery for Morbid Obesity

**DOI:** 10.3390/jcm9113430

**Published:** 2020-10-26

**Authors:** Karl Peter Rheinwalt, Uta Drebber, Robert Schierwagen, Sabine Klein, Ulf Peter Neumann, Tom Florian Ulmer, Andreas Plamper, Andreas Kroh, Sandra Schipper, Margarete Odenthal, Frank Erhard Uschner, Philipp Lingohr, Jonel Trebicka, Maximilian Joseph Brol

**Affiliations:** 1Department of Bariatric, Metabolic, and Plastic Surgery, St. Franziskus-Hospital, 50825 Cologne, Germany; karlpeter.rheinwalt@cellitinnen.de (K.P.R.); andreas.plamper@cellitinnen.de (A.P.); 2Department of Pathology, University Hospital of Cologne, University of Cologne, 50937 Cologne, Germany; uta.drebber@uk-koeln.de (U.D.); margarete.odenthal@uk-koeln.de (M.O.); 3Translational Hepatology, Department of Internal Medicine I, University Clinic Frankfurt, 60590 Frankfurt, Germany; Robert.Schierwagen@kgu.de (R.S.); Sabine.Klein@kgu.de (S.K.); Frank.Uschner@kgu.de (F.E.U.); brol@uni-bonn.de (M.J.B.); 4Clinic for General, Visceral and Transplantation Surgery, University Hospital RWTH Aachen, 52074 Aachen, Germany; uneumann@ukaachen.de (U.P.N.); fulmer@ukaachen.de (T.F.U.); akroh@ukaachen.de (A.K.); sschipper@ukaachen.de (S.S.); 5Department of Surgery, Maastricht University Medical Centre (MUMC), P.O. Box 5800, 6202 AZ Maastricht, The Netherlands; 6Department of Nanomedicine and Theranostics, Institute for Experimental Molecular Imaging, RWTH University Clinic and Helmholtz Institute for Biomedical Engineering, 52074 Aachen, Germany; 7Department of General, Visceral, Thoracic and Vascular Surgery, University Hospital Bonn, 53127 Bonn, Germany; Philipp.lingohr@ukbonn.de; 8European Foundation for the Study of Chronic Liver Failure-EF Clif, 08021 Barcelona, Spain; 9Department of Internal Medicine I, University Clinic, 53127 Bonn, Germany

**Keywords:** Roux-en-Y gastric bypass, one anastomosis gastric bypass, mini gastric bypass, NAFLD, NASH, weight loss

## Abstract

Background. Bariatric surgery is a widely used treatment for morbid obesity. Prediction of postoperative weight loss currently relies on prediction models, which mostly overestimate patients’ weight loss. Data about the influence of Non-alcoholic fatty liver disease (NAFLD) on early postoperative weight loss are scarce. Methods. This prospective, single-center cohort study included 143 patients receiving laparoscopic gastric bypass surgery (One Anastomosis-Mini Gastric Bypass (OAGB-MGB) or Roux-en-Y Gastric Bypass (RYGB)). Liver biopsies were acquired at surgery. NAFLD activity score (NAS) assigned patients to “No NAFLD”, “NAFL” or “NASH”. Follow up data were collected at 3, 6 and 12 months. Results. In total, 49.7% of patients had NASH, while 41.3% had NAFL. Compared with the No NAFLD group, NAFL and NASH showed higher body-mass-index (BMI) at follow-up (6 months: 31.0 kg/m^2^ vs. 36.8 kg/m^2^ and 36.1 kg/m^2^, 12 months: 27.0 kg/m^2^ vs. 34.4 and 32.8 kg/m^2^) and lower percentage of total body weight loss (%TBWL): (6 months: 27.1% vs. 23.3% and 24.4%; 12 months: 38.5% vs. 30.1 and 32.6%). Linear regression of NAS points significantly predicts percentage of excessive weight loss (%EWL) after 6 months (Cologne-weight-loss-prediction-score). Conclusions. Histopathological presence of NAFLD might lead to inferior postoperative weight reduction after gastric bypass surgery. The mechanisms underlying this observation should be further studied.

## 1. Introduction

The obesity pandemic is responsible for a broad variety of obesity-related comorbidities. These comorbidities such as diabetes mellitsus type 2, arterial hypertension, obstructive sleep apnea syndrome, fatty liver disease, pathologies of the musculoskeletal system, and obesity-associated malignancies are causing an increasing health care burden [1]. Moreover, in obese subjects, an increased waist-to-hip ratio is positively associated with coronary atherosclerotic disease [2]. Weight reduction is the only approach to achieve remission and potentially cure obesity-related comorbidities. Therefore, weight loss in obese individuals is of pivotal interest. In daily clinical practice, conservative approaches alone, such as particular nutritional regimens, physical exercise, and pharmaceutical treatments are frequently insufficient [3]. As ultima ratio, bariatric surgery is performed, since it is clinically and economically effective and safe, mainly due to improvement in laparoscopic techniques. It is possibly underutilized since less than 1% of obese adults receive this treatment [4,5].

Weight loss after bariatric surgery is difficult to predict. In bypass patients with postoperative persisting hunger as a sign of maladaptation to the enterohormonal changes, genetic factors might play a causal role [6]. Current prediction models for weight loss after bariatric surgery overestimate the outcome and are therefore not routinely used in clinical practice [7]. Neural factors, preoperative weight loss, and early postoperative weight loss were identified as predictors of the magnitude of postoperative weight loss [8,9,10]. However, specific clinical phenotypes to predict postoperative weight loss remain unknown.

Non-alcoholic fatty liver disease (NAFLD) includes a wide range of pathological alterations in metabolism, muscle and adipose tissue homeostasis with variations depending on its severity. This is due to a multiple hit pathogenesis including genetic, dietary and gut microbial factors [11]. With a global prevalence of about 25% NAFLD is highly associated with obesity [12]. The spectrum covers simple steatosis/nonalcoholic fatty liver (NAFL), nonalcoholic steatohepatitis (NASH) with or without fibrosis and NASH cirrhosis. Recent studies showed specific serum RNA profiles in severe and mild NAFLD, possibly mirroring molecular mechanisms responsible for the progression towards NASH [13]. The impact of NAFLD on postoperative weight loss after bariatric surgery has not been investigated in detail. In accordance with recent findings of other study groups, we hypothesized that the presence of NAFLD alters post-operative weight loss [14]. The current study evaluates hepatic histomorphology as a potential predictor for postoperative weight loss after gastric bypass surgery.

## 2. Materials and Methods

### 2.1. Study Design

This prospective, single-center, non-interventional and longitudinal study was performed at the Department of Bariatric, Metabolic and Plastic Surgery, St. Franziskus-Hospital Cologne, Germany, between July 2018 and May 2019. The clinic is a tertiary care center performing around 350 primary and 100 bariatric revision-procedures per year.

Informed consent was obtained from all patients prior to inclusion. The study was conducted in accordance with the Declaration of Helsinki, and the protocol was approved by the ethics committees of the regional Medical Association (Nordrhein) (project identification code 2017110) and the University of Bonn (project identification code 194/17).

### 2.2. Study Population

Figure 1 presents the patient flow chartof the study. All patients planned for either One Anastomosis/Mini-gastric bypass (OAGB-MGB) or Roux-en-Y gastric bypass (RYGB) were screened for inclusion during the study period. Exclusion criteria were age below 18 years, alcohol consumption above one standard drink per day, presence of hepatopathy of any other etiology than metabolic and clinical and/or histopathological evidence of liver cirrhosis, previous bariatric surgery, surgical revisions and/or conversion procedures. A total of 143 patients receiving primary bariatric bypass surgery as treatment for morbid obesity were included. Data on comorbidities were collected by standardized interviews. Musculoskeletal disorders must not have been caused by trauma or an autoimmune disease and required medical, physiotherapeutic and/or surgical treatment. Reasons for non-participation were not recorded. Follow-up was conducted every three months for a total of 12 months. Drop-out rates were 3.5% after three months, 4.2% after six months and 39.9% after 12 months.

### 2.3. Standard Laboratory Values and NAFLD Scores at Baseline

Standard laboratory values (white and red blood cell count, platelets, international normalized ratio (INR), electrolytes, creatinine, urea, transaminases, γ-glutamyltransferase, bilirubin, total serum protein and albumin) were assessed at baseline in order to calculate previously published biomarker scores for every patient. Body mass index (BMI) was defined as weight in kilogram divided by the square of the body height in meters. In detail, we calculated the BARD score for NAFLD fibrosis, which contains three variables in a weighted sum (BMI > 28 kg/m^2^ = 1 point, presence of diabetes mellitus = 1 point, aspartate aminotransferase (AST)/alanine aminotransferase (ALT) ratio > 0.8 = 2 points) [15], the AST to platelet ratio index [16], the fibrosis-4 (FIB-4) index for liver fibrosis [17] as well as the NAFLD fibrosis score [18].

### 2.4. Liver Biopsy

Intraoperative liver biopsy was performed immediately after trocar placement by taking a wedge biopsy of approximately 1 × 1 × 1 cm of segment III. Specimens were sent to the Department for Pathology, University of Cologne. Liver biopsies were processed according to standard protocol. Fixation was performed overnight in 4% buffered formalin and embedded in paraffin. Four micrometer sections were stained with hematoxylin and eosin (HE), van Gieson stain, periodic acid Schiff after diastase, Gomori and Prussian blue.

### 2.5. Histopathological Evaluation

The histological evaluation was performed by applying the “Nonalcoholic Fatty Liver Disease Activity Score System” from the Nonalcoholic Steatohepatitis Clinical Research Network Study Group (NASH-CRN) [19] blinded by two expert pathologists with a high concordance of the results (UD, MO). In this system, the histological criteria of fat, ballooning degeneration and lobular inflammation are quantified as follows: fat: ≤ 5% score 0, 6%–33% score 1, 34%–66% score 2, 67%–100% score 3; ballooning degeneration: none score 0, few score 1, many score 2; lobular inflammation: none score 0, <2 foci per 20x field score 1, 2–4 foci per 20x field score 2, >4 foci per 20x field score 3.

Fibrosis staging: 1a mild pericellular fibrosis; only seen in trichrome stain, 1b moderate pericellular fibrosis; readily seen on HE stain, 1c only portal fibrosis with no pericellular fibrosis, 2 portal fibrosis (any) and pericellular fibrosis (any), 3 bridging fibrosis, 4 cirrhosis.

According to this system, biopsies with a total score of 1or2 are diagnosed as NAFL; cases with score 5 or higher are diagnosed as NASH. Biopsies with activity scores of 3 or 4 can be either NAFL or NASH.

### 2.6. Statistical Analysis

Descriptive statistics were computed for all variables. Categorical variables were reported as absolute frequency (*n*) and percentage (%). Continuous variables were reported as median and interquartile range (IQR). Included patients were divided into three groups according to their liver diagnosis (no NAFLD, NAFL, NASH). For categorical variables, Chi-square tests were used in univariate analyses. Nonparametric tests were used for comparisons between groups, if applicable. For continuous variables, Kruskal–Wallis test was used to check for differences among the three groups. Pairwise comparisons were performed using Mann–Whitney U tests. Multiple linear regression analysis was used to develop a model for postoperative weight loss prediction. *p* values below 0.05 were considered statistically significant. Data were analyzed using SPSS Statistics (Version 26, IBM, Armonk, NY, USA). Graphs were performed using Prism V.5.0 (GraphPad, San Diego, CA, USA). Data were expressed as mean ± standard error of the mean (SEM), unless otherwise specified.

Apart from assessment of percentage of excessive weight loss (%EWL), we calculated percentage of total body weight loss (%TBWL) for all individuals, since this parameter has become widely accepted as being more robust and not dependent on pre-operative overweight [20,21].

## 3. Results

A total of 143 patients with gastric bypass operation were included in the study. Based on histopathological NAS activity score in the liver biopsy, individuals were grouped into three groups: “No NALFD”, “NAFL” and “NASH”. Figure 2A shows representative HE staining for each group.

### 3.1. Demographics and Descriptive Analysis

The patient characteristics are detailed in Table 1. According to histopathological assignment, three groups were identified: “No NAFLD” (*n* = 13, 9.1%), “NAFL” (*n* = 59, 41.3%) and “NASH” (*n* = 71, 49.7%). Median age was 42 (34–50) years, median BMI was 49.3 (44.6–55.1) kg/m^2^. More women (79.7%) than men were included in the study. In total, 62.2% of patients had preoperative arterial hypertension, predominantly patients with NASH. Almost all patients had musculoskeletal disorders, whereas only one patient suffered from coronary heart disease. Nearly every second patient receiving bariatric surgery showed histological evidence of NASH. Interestingly, nobody in the “No NAFLD” group had type 2 diabetes, while 27.1% in the “NAFL” and 47.9% in the “NASH” group were diagnosed as diabetics.

No surgical- and/or biopsy-related perioperative complications occurred.

### 3.2. Baseline AST Platelet Index Identifies NASH in Patients Undergoing Bariatric Surgery

Median and IQR for liver biomarker scores are displayed in Table 2. We calculated common biomarker scores of liver diseases in order to evaluate which of our three groups fits best for preoperative stratifying. AST platelet index varied significantly between the groups, with lowest values in the “No NAFLD” group, followed by NAFL and highest values in the NASH group (Figure 2B). BARD score was similar among the three groups. While FIB-4 score did not differ significantly between the groups, it was nevertheless lower in NALFD patients (Table 2). Similar results were found for the NAFLD fibrosis score, rendering these scores not useful for proper preoperative stratification.

### 3.3. NAFLD and NASH Led to Inferior Weight Loss after Gastric Bypass Surgery

Evolution of BMI, BMI loss, %TBWL and %EWL are shown in Figure 3A–D. At baseline, median BMI did not vary among the three groups (*p* = 0.531) (Table 1). Follow-up data are shown in Table 3.

At three months follow-up, patients in the “No NALFD” group had a median BMI of 35.5 kg/m^2^. Patients with NAFL or NASH had a statistically higher BMI (41.2 kg/m^2^, *p* = 0.020 and 40.9 kg/m^2^, *p* = 0.026 respectively) compared to patients in the “No NAFLD” group (Figure 3A). However, loss of BMI did not show a significant difference between the three groups (Figure 3B).

Furthermore, at three months follow-up (Table 3), patients in the “No NAFLD” group performed best according to %TBWL with 18.1% (Figure 3C). Patients in the NAFL and NASH group had %TBWL of 16.5 and 16.8% respectively. Patients with NAFL lost on average significantly less %TBWL than patients with “No NAFLD”.

After six months, differences were most pronounced. Six months after the bariatric procedure, BMI was significantly higher, %TBWL and %EWL were significantly less in the NAFL and NASH group compared to the “No NAFLD” group (Figure 3C,D). Interestingly, BMI loss was not affected among these groups.

Finally at the annual follow-up after 12 months, BMI in the “no NAFLD” group was significantly lower than in patients with NAFL. %TBWL was significantly lower in the NASH group. Patients with NAFL had the lowest median %TBWL, while %EWL was significantly higher in the “No NAFLD” group compared to both NAFL and NASH patients (Table 3).

In summary, patients with NAFLD, whether NAFL or NASH, had lost less weight one year after gastric bypass surgery. The most pronounced effects were observed six months after gastric bypass surgery.

### 3.4. Multiple Regression Model for Prediction of %EWL after 6 Months

Next, we assessed for the possibility of weight loss prediction using clinical, laboratory and histopathological data. Since clinical and laboratory data as well as the scores performed poorly, NAS points for steatosis, ballooning and inflammation at baseline histology were combined in a multiple linear regression model to investigate the potential of a weight loss prediction model. These variables resulted in a model which predicts the percentage of excessive weight loss (%EWL) after six months: (58.189 + (0.477 × NAS-Steatosis)–(3.875 × NAS-Ballooning)–(1.313 × NAS-Inflammation)) (r = 0.239, *p* = 0.049). This score was named “Cologne-weight-loss-prediction-score” and is available as an online calculator at http://www.stfranziskus.de/medizin/kliniken/chirurgie-iii-adipositas-metabolische-und-plastische-chirurgie/.

## 4. Discussion

Bariatric surgery is known to be an effective treatment for morbid obesity and associated comorbidities. Bedossa et al. found that prevalence of NAFLD in bariatric surgery patients was as high as 78% in 798 cases, which roughly corresponds to our finding of 90.9% [22].

This is the first study showing that baseline liver histology is significantly associated with post-operative weight loss. We could demonstrate that histological presence of NAFL or NASH at surgery was associated with decreased weight loss three, six and 12 months after OAGB-MGB or RYGB. We further developed a biopsy-based model, which may predict %EWL weight loss after six months. Many studies provided evidence that bariatric surgery is improving NAFLD, indicating that these subjects benefit from a gastric bypass procedure. Improvement of NAFL/NASH seems to be best demonstrated for RYGB [23], particularly with better results compared to purely restrictive weight loss surgery such as adjustable gastric banding [24]. Recently, a French single-center study reported an association between the persistence of NASH after bypass surgery and lower weight loss [14]. This study provided a second biopsy after five years. However, only patients with preexisting NASH were included and therefore the authors failed to compare the weight–loss results of patients with and without NASH. Our study relies on a baseline biopsy investigating its impact on postoperative weight loss. Indeed, the wide IQR of our NAFLD patients at 12 months may be an indirect sign of a different course of NAFLD resolution after bariatric surgery, possibly confirming the previously published data.

Many attempts have been made to determine baseline predictive factors for early weight loss after bariatric surgery. Preoperative caloric intake and bioelectrical impedance analysis have been described to predict weight loss after bariatric surgery [25,26]. Moreover, it was demonstrated for RYGB that higher baseline BMI leads to increased postoperative %EWL [27]. Predictive models for weight loss rely on clinical and laboratory data, but to date, the presence of liver comorbidity has not been taken into consideration, presumably due to obstacles in gaining preoperative liver biopsies [7]. Yet, presence of NAFLD seems to be clearly associated with metabolism, muscle and adipose tissue homeostasis, thereby determining body weight evolution after surgery.

Pre-operative scores using non-invasive biomarkers must be interpreted carefully. These scores were originally designed and validated for other purposes, such as fibrosis, hepatitis C virus progression or mortality [15,16,17,18]. In our research, these scores showed no usefulness in predicting weight loss, whereas baseline histology of liver disease could stratify patients. Since stratification and diagnosis of NALFD through liver biopsy remains gold-standard [28] and due to its high (>90%) prevalence in this population as shown in our study, and its potential role on early weight loss, it is expedient to recommend routine liver biopsy at surgery.

Genetic factors can further explain some of the weight loss variability after gastric bypass surgery. A recent study provided a genetic risk score, based on single nucleotide polymorphisms, to increase the accuracy of the predicted postoperative weight reduction [29]. Moreover, postoperative weight loss is mostly driven by enterohormonal changes, involving Ghrelin, glucagon like peptide-1, cholecystokinin and peptide YY(1–36) [30]. Secretion of these hormones changes towards an anorectic profile after bypass surgery [31]. Therefore, persistent early postoperative hunger may additionally alter postoperative weight loss. Both of these interesting mechanisms need further exploration but were beyond the scope of our study. At the same time, the exact mechanisms of weight loss linked to NAFLD remains to be established.

Despite the lesser weight loss in the NASH group, these individuals greatly benefit from gastric bypass intervention. In a very recently published single-center study including 180 biopsy-proven NASH patients who underwent bariatric surgery, 84% of the study cohort showed NASH resolution after five years and 70% had a reduction of liver fibrosis [14]. Another study was able to show an association between an improvement in preoperatively impaired liver function due to NASH and weight loss after bariatric surgery [32]. In this study, liver function was measured by the LiMAx^®^ test (enzymatic capacity of cytochrome P4501A2) [33]. These results indicate that bariatric procedures can be used as NASH treatment. However, the important question remains as to whether pre- or intraoperative biopsy should be performed routinely in the context of gastric bypass surgery. Our study advocates for the use of intraoperative liver biopsy, which is useful in the prediction of weight loss, and available to the public at http://www.stfranziskus.de/medizin/kliniken/chirurgie-iii-adipositas-metabolische-und-plastische-chirurgie/ (remains to be established after acceptance for publication).

Our study has several limitations. The small number of individuals in the “No NAFLD” group is one drawback of this study. However, the high prevalence (>90%) of NAFLD among morbidly obese patients (BMI > 40 kg/m^2^) requiring metabolic surgery is a fact, which also reflects a real-life scenario in this population and an important take-home message. Furthermore, this potential selection bias may be reduced by blinded read-outs. The blinded read-outs of the liver biopsies allowed an unbiased assignment of our cohort. A further limitation is that only half of the patients received a 12-months follow-up visit, enabeling only cautious prediction for long-term weight loss, which may be explained by the different course of NAFLD in these patients as reflected by the large IQR.

Moreover, further variables possibly influencing weight loss were not investigated in the present study. For example, alterations of the microbiome have been described to have a varying impact on remission of diabetes mellitus after bariatric procedures. Most abundant changes were identified in patients who underwent RYGB [34].

## 5. Conclusions

Our prospective cohort study provides first evidence that patients with metabolic liver disease (NAFLD and NASH) have lower weight loss after RYGB and OAGB-MGB than patients without NAFLD. Therefore, we recommend baseline biopsies at surgery to guide management of patients and to provide an easy-to-use prediction score at surgery.

## Figures and Tables

**Figure 1 jcm-09-03430-f001:**
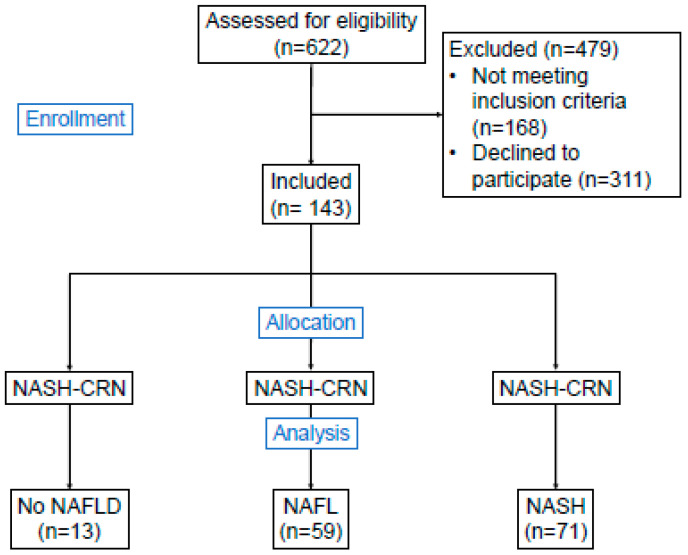
Study flowchart. Study flowchart showing amount of patients assessed for eligibility, number of patients not enrolled and allocation according to histopathology and the number of patients in the three populations (no NAFLD, NAFL and NASH). Allocation was performed by using the histological NAFLD activity score from the NASH-CRN. Abbreviations: NAFLD—Non-alcoholic fatty liver disease; NAFL—Non-alcoholic fatty liver; NASH—Non-alcoholic steatohepatitis; NASH-CRN—Nonalcoholic Steatohepatitis Clinical Research Network Study Group.

**Figure 2 jcm-09-03430-f002:**
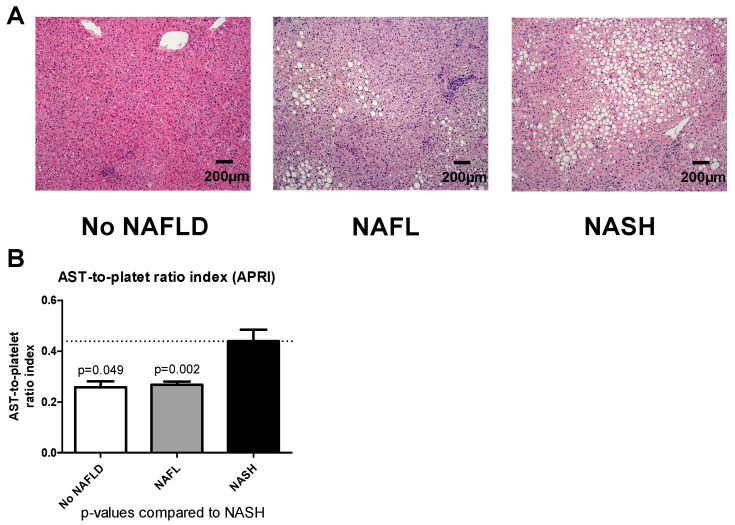
Histological stainings and APRI Representative hematoxylin and eosin stainings (**A**) of patients in the “No NAFLD”, “NAFL” and “NASH” group. No NAFLD is characterized by absence of relevant steatosis, ballooning, inflammatory reaction. NAFL patients showed modest steatosis, ballooning, and inflammatory reaction. NASH patients showed severe steatosis, ballooning, inflammation. Slides are captured at a 100x magnification, scale bars are 200 µm. Graph showing APRI for “No NAFLD”, “NAFL” and “NASH” (**B**). Data are expressed as mean ± standard error of the mean (SEM). *p*-values for comparisons with the “NASH” group are provided above the bar. Abbreviations: NAFLD—non-alcoholic fatty liver disease; NAFL—non-alcoholic fatty liver; NASH—non-alcoholic steatohepatitis; APRI—AST-to-platelet ratio index.

**Figure 3 jcm-09-03430-f003:**
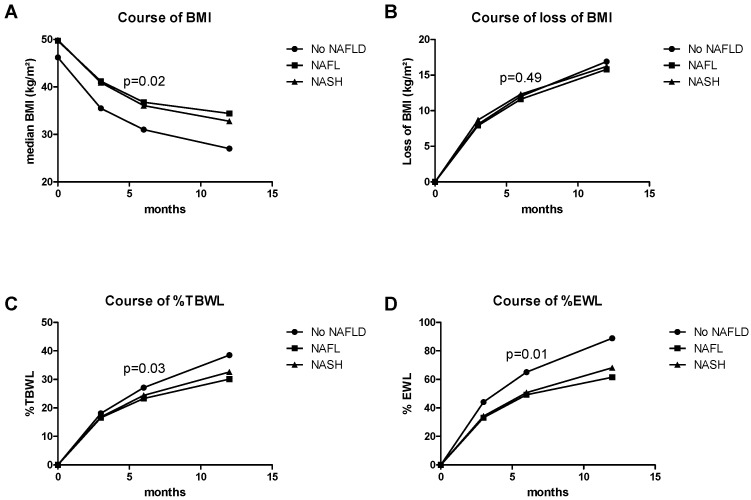
Evolution of weight loss, BMI loss, %TBWL and %EWL. Diagrams showing the evolution of BMI (**A**), BMI loss (**B**), %TBWL (**C**) and %EWL (**D**) at baseline and after 3, 6 and 12 months postoperatively for “No NAFLD”, “NAFL” and “NASH” patients. Data are plotted as medians. *p*-values within the graph were computed for the six months follow-up visit by Kruskal–Wallis test. Abbreviations: BMI—body mass index; %TBWL—percentage of total body weight loss; %EWL—percentage of excessive weight loss.

**Table 1 jcm-09-03430-t001:** Characteristics rrior to enrollment, in patients with “non-steatotic liver” (No NAFLD), “non-alcoholic fatty liver (NAFL)” and “non-alcoholic steatohepatitis” (NASH).

Characteristic	No NAFLD (*n* = 13)	NAFL (*n* = 59)	NASH (*n* = 71)	*p* Value
Data at baseline				
Age, y, median (IQR)	41 (33–49)	41 (33–52)	43 (35–50)	0.204
Female sex, *n* (%)	11 (84.6)	46 (78.0)	57 (80.3)	0.853
Body mass index, kg/m^2^, median (IQR)	46.2 (41.1–49.1)	49.7 (44.8–55.4)	49.4 (44.6–55.2)	0.531
Obesity-related comorbidities, *n* (%)				
Arterial hypertension	5 (38.5)	35 (59.3)	49 (69.0) ^a^	0.094
Obstructive sleep apnea syndrome	6 (46.2)	45 (76.3) ^a^	47 (66.2)	0.089
Coronary heart disease	0 (0.0)	0 (0.0)	1 (1.4)	0.600
Type 2 diabetes mellitus	0 (0.0)	16 (27.1) ^a^	34 (47.9) ^a,b^	0.001
Musculoskeletal disorder	13 (100.0)	59 (100.0)	69 (97.2)	0.358
Type of bariatric surgery, *n* (%)				
One Anastomosis/Mini-Gastric Bypass (OAGB-MGB)	11 (84.6)	51 (86.4)	67 (94.4)	0.247
Duration of bariatric surgery, min, median (IQR)	70 (62–79.5)	80 (68–90)	75 (67–90)	0.304
Duration of postoperative hospitalization, days, median (IQR)	3 (3–3)	3 (3–4)	3 (3–3)	0.733

*p* values were obtained using chi-square test or Kruskal–Wallis Test. ^a^ Significantly different from the “No NAFLD” group. ^b^ Significantly different from the “NAFL” group.

**Table 2 jcm-09-03430-t002:** Biomarker scores and laboratory findings at enrollment in patients with “non-steatotic liver” (No NAFLD), “non-alcoholic fatty liver (NAFL)” and “non-alcoholic steatohepatitis” (NASH).

Characteristic	No NAFLD (*n* = 13)	NAFL (*n* = 59)	NASH (*n* = 71)	*p* Value
Biomarker scores prior to bariatric surgery, median (IQR)				
BARD score	3 (1.5–3)	3 (1–3)	3 (2–3)	NA
AST platelet ratio index	0.24 (0.17–0.31)	0.25 (0.19–0.31)	0.30 (0.21–0.52) ^b^	0.006
FIB-4 score	0.72 (0.59–1.02)	0.67 (0.49–0.86)	0.80 (0.59–1.08)	0.102
NAFLD fibrosis score	−0.77 (−1.39–0.26)	−0.87 (−1.59–0.17)	−0.63 (−1.17–0.11)	0.373
Laboratory findings median (IQR)				
White-cell count, × 10^9^/L	7.3 (6.7–8.7)	7.7 (6.6–9.1)	7.9 (6.8–8.7)	0.784
Hemoglobin, g/dL	13.4 (12.3–14.3)	14 (13.1–14.7)	14.0 (13.4–15.1)	0.172
Platelet count, × 10^9^/L	250 (230–305)	287 (245–334)	293 (237–324)	0.278
Serum C-reactive protein, mg/L	1.2 (0.6–1.7)	0.8 (0.4–1.4)	1.2 (0.6–1.6)	0.214
Serum bilirubin, mg/dL, median (IQR)	0.5 (0.5–0.6)	0.6 (0.5–0.7)	0.6 (0.5–0.8)	0.115
Serum albumin, g/dL	40.3 (39.5–42.9)	41.6 (40.4–43.4)	42.2 (40.4–43.8)	0.141
International normalized ratio	1.01 (0.97–1.09)	1.01 (0.98–1.03)	1.01 (0.98–1.04)	0.767
Serum creatinine, mg/dL	0.9 (0.8–0.9)	0.9 (0.8–1.0)	0.9 (0.8–1.0)	0.377

*p* values were obtained using Kruskal–Wallis Test, ^b^ Significantly different from the “NAFL” group.

**Table 3 jcm-09-03430-t003:** Characteristics three, six and 12 months after bariatric surgery, in patients with “non-steatotic liver” (No NAFLD), “non-alcoholic fatty liver (NAFL)” and “non-alcoholic steatohepatitis” (NASH).

Characteristic	No NAFLD	NAFL	NASH	*p* Value
Data at 3 months follow-up	(*n* = 13)	(*n* = 56)	(*n* = 69)	
Body mass index, kg/m^2^	35.5 (32.9–42.1)	41.2 (37.7–45.4) ^a^	40.9 (37.3–46.3) ^a^	0.057
Loss of body mass index, kg/m^2^	8.1 (7.5–9.9)	7.9 (6.9–9.2)	8.7 (7.0–9.6)	0.491
Total body weight loss, %	18.1 (16.4–19.9)	16.5 (14.2–18.0) ^a^	16.8 (14.7–19.6)	0.068
Excessive weight loss, %	44.1 (31.6–51.4)	33.1 (27.8–39.2) ^a^	34.3 (28.9–39.9) ^a^	0.037
Data at 6 months follow-up	(*n* = 12)	(*n* = 57)	(*n* = 68)	
Body mass index, kg/m^2^	31.0 (28.7–36.8)	36.8 (34.2–41.4) ^a^	36.1 (33.9–41.4) ^a^	0.020
Loss of body mass index, kg/m^2^	12.0 (11.0–14.5)	11.6 (10.0–13.5)	12.3 (10.1–14.3)	0.492
Total body weight loss, %	27.1 (25.8–29.7)	23.3 (20.0–26.9) ^a^	24.4 (21.1–27.2) ^a^	0.026
Excessive weight loss, %	65.1 (47.7–77.9)	49.2 (43.6–55.5) ^a^	50.7 (41.8–59.0) ^a^	0.008
Data at 12 months follow-up	(*n* = 6)	(*n* = 39)	(*n* = 41)	
Body mass index, kg/m^2^	27.0 (23.9–31.2)	34.4 (30.4–38.1) ^a^	32.8 (29.7–37.6) ^a^	0.040
Loss of body mass index, kg/m^2^	16.9 (15.7–21.0)	15.8 (13.3–20.2)	16.2 (13.3–20.8)	0.736
Total body weight loss, %	38.5 (34.9–42.7)	30.1 (27.4–38.3)	32.6 (26.4–40.4) ^a^	0.113
Excessive weight loss, %	88.9 (75.4–107.0)	61.5 (54.1–75.3) ^a^	68.1 (53.5–77.4) ^a^	0.180

*p* values were obtained using Kruskal-Wallis test. ^a^ Significantly different from the “No NAFLD” group. *p* < 0.05 were considered significantly different.

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
