# Peer review of "Baseline Presence of NAFLD Predicts Weight Loss after Gastric Bypass Surgery for Morbid Obesity"

_jcm, 2020, doi:10.3390/jcm9113430_

Round 1

Reviewer 1 Report

In this manuscript, Rheinwalt et al. focused on weight lose by 143 patients receiving laparoscopic gastric bypass surgery (OAGB-MGB  or  RYGB).  The aim is scientifically and clinically relevant. Authors showed that patients with metabolic liver disease (NAFLD and NASH) have lower weight loss after bariatric surgery than patients without NAFLD. Authors recommend baseline biopsies at surgery to guide management of patients and offer an easy-to-use score to predict it at surgery.

 Comments:

The introduction is interesting and provides enough background. Several edits are required along with the text:

-  please define all abbreviations (e.g. Ln 110 and 200: AST, ALT, TBWL, EWL);

- please use the same notation for “p” in figure (e.g.: p= .0485), in tables (e.g.: p=0.001) and in text;

While “p” is often italicized, the values are typically not. You can updated all p-values accordingly.

- please clarify the BMI formula in the text;

- please graphically correct figure 2A - scale are unreadable;

- please graphically correct figure 2B - e.g. add the dotted horizontal line represents the mean value for NASH.

Author Response

Response to Reviewer 1

In this manuscript, Rheinwalt et al. focused on weight lose by 143 patients receiving laparoscopic gastric bypass surgery (OAGB-MGB  or  RYGB).  The aim is scientifically and clinically relevant. Authors showed that patients with metabolic liver disease (NAFLD and NASH) have lower weight loss after bariatric surgery than patients without NAFLD. Authors recommend baseline biopsies at surgery to guide management of patients and offer an easy-to-use score to predict it at surgery.

We thank the reviewer for her/his kind consideration of our work.

Comments:

The introduction is interesting and provides enough background. Several edits are required along with the text:

-  please define all abbreviations (e.g. Ln 110 and 200: AST, ALT, TBWL, EWL);

We appreciate this comment. All abbreviations are now introduced when they appear for the first time in the manuscript.

- please use the same notation for “p” in figure (e.g.: p= .0485), in tables (e.g.: p=0.001) and in text;

While “p” is often italicized, the values are typically not. You can updated all p-values accordingly.

Thank you for this comment. We standardized the notation for “p”: It appears now non-italic with a leading zero throughout the manuscript, the tables and the figures. (e.g.: p=0.0485).

- please clarify the BMI formula in the text;

We provided the definition of BMI in the Materials and Method section: “Body mass index (BMI) was defined as weight in kilogram divided by the square of the body height in meters.”

- please graphically correct figure 2A - scale are unreadable;

We thank the reviewer for this important comment. We have enlarged the scale bar legends and harmonized them to 200µm for each slide.

- please graphically correct figure 2B - e.g. add the dotted horizontal line represents the mean value for NASH.

We graphically corrected figure 2B: p-values were harmonized according to the tables and the manuscript, and a dotted horizontal line for the mean value for NASH was added.

Reviewer 2 Report

Fig. 1, please describe the meaning of CRN also in the figure legend

Pag. 4, describe in more  details the BARD score composition and its calculation algorithm

Pag. 5. provide more details on the type and diagnostic workflow of muscoloskeletal disorders

Page 7 replace the expression "statistically significantly" with only "significantly" (only one adverb)

In general, I would underline the fact that the higher baseline BMI among the 3 populations influences the 6 and 12 months weight loss as the % progressive decrease does not differ in a significant fashion. 

Author Response

Response to Reviewer 2

Fig. 1, please describe the meaning of CRN also in the figure legend

We thank the reviewer for his/her comment. We added the following sentence to the figure legend in Fig. 1: “Allocation was performed by using the histological NAFLD activity score from the NASH-CRN.”

Pag. 4, describe in more  details the BARD score composition and its calculation algorithm

We are grateful for this comment. We precised the calculation of the BARD in the Materials and methods section: “In detail, we calculated the BARD score for NAFLD fibrosis, which contains three variables in a weighted sum (BMI>28kg/m² = 1 point, presence of diabetes mellitus = 1 point, aspartate aminotransferase(AST)/alanine aminotransferase(ALT) ratio> 0.8 = 2 points)12.”

Pag. 5. provide more details on the type and diagnostic workflow of muscoloskeletal disorders

Screening for musculoskeletal disorders utilized data from patients´ history, diagnostic, and therapy. One or more of the following findings affirmed this immobilizing comorbidity: repetitive or chronic pain at the level of the lower spine and/or lower extremities with absence of trauma and/or autoimmune disease as principal cause, presence of severe osteoarthrosis on imaging studies, need for repetitive orthopedic and/or medical therapy (non-steroidal anti-inflammatories or morphins), joint replacement or reconstructive spinal surgery indicated or already realized. We added information in the Materials and methods section.

Page 7 replace the expression "statistically significantly" with only "significantly" (only one adverb)

As suggested by the reviewer, this has been adjusted.

In general, I would underline the fact that the higher baseline BMI among the 3 populations influences the 6 and 12 months weight loss as the % progressive decrease does not differ in a significant fashion. 

Thank you for this comment. We added the fact that baseline BMI influences weight loss at 6 and/or 12 months after surgery in the discussion and added reference 27: 10.1016/j.soard.2018.05.014. However, Kruskal-Wallis-Test analysis did not show a significant difference for baseline BMI in our cohort in contrast to the 6 and 12 month follow-up.

Reviewer 3 Report

In this manuscript the authors found that histopathological presence of NAFLD might lead to inferior postoperative weight reduction after gastric bypass surgery. The aim is clear and the discussion is well performed; however, I have some suggestions.

  • In the Introduction Section, the authors commented the comorbidities associated to obesity but I don't see cardiovascular disease, especially coronary heart disease; please consider this reference (10.1007/s00592-018-1144-9) and comment this in the Introduction.
  • In the Introduction Section, the authors did not focuse on  fatty liver pathophysiological players; please consider these references (10.1111/liv.14167; 10.3390/ijms20081948; 10.1016/j.numecd.2014.01.013) and comment these in the Introduction.

Author Response

Response to Reviewer 3

In this manuscript the authors found that histopathological presence of NAFLD might lead to inferior postoperative weight reduction after gastric bypass surgery. The aim is clear and the discussion is well performed; however, I have some suggestions.

We are very grateful for the kind consideration of our work.

In the Introduction Section, the authors commented the comorbidities associated to obesity but I don't see cardiovascular disease, especially coronary heart disease; please consider this reference (10.1007/s00592-018-1144-9) and comment this in the Introduction.

Thank you for this comment. As suggested, we included the reference 10.1007/s00592-018-1144-9 in the introduction section and pointed out that obese subjects with an increased waist-to-hip ratio are at risk to develop coronary heart disease.

In the Introduction Section, the authors did not focuse on  fatty liver pathophysiological players; please consider these references (10.1111/liv.14167; 10.3390/ijms20081948; 10.1016/j.numecd.2014.01.013) and comment these in the Introduction.

We appreciate this comment. In the revised manuscript, we cited and discussed the references 10.1111/liv.14167 and 10.3390/ijms20081948 and added further information about the pathophysiological players involved in NAFLD pathogenesis.
